# Molecularly Imprinted Polymers as State-of-the-Art Drug Carriers in Hydrogel Transdermal Drug Delivery Applications

**DOI:** 10.3390/polym14030640

**Published:** 2022-02-08

**Authors:** Aleksandra Lusina, Michał Cegłowski

**Affiliations:** Faculty of Chemistry, Adam Mickiewicz University in Poznań, Uniwersytetu Poznańskiego 8, 61-614 Poznań, Poland

**Keywords:** Molecular Imprinted Polymers (MIP), Molecular Imprinting Technology (MIT), hydrogels, transdermal drug delivery

## Abstract

Molecularly Imprinted Polymers (MIPs) are polymeric networks capable of recognizing determined analytes. Among other methods, non-covalent imprinting has become the most popular synthesis strategy for Molecular Imprinting Technology (MIT). While MIPs are widely used in various scientific fields, one of their most challenging applications lies within pharmaceutical chemistry, namely in therapeutics or various medical therapies. Many studies focus on using hydrogel MIPs in transdermal drug delivery, as the most valuable feature of hydrogels in their application in drug delivery systems that allow controlled diffusion and amplification of the microscopic events. Hydrogels have many advantages over other imprinting materials, such as milder synthesis conditions at lower temperatures or the increase in the availability of biological templates like DNA, protein, and nucleic acid. Moreover, one of the most desirable controlled drug delivery applications is the development of stimuli-responsive hydrogels that can modulate the release in response to changes in pH, temperature, ionic strength, or others. The most important feature of these systems is that they can be designed to operate within a particular human body area due to the possibility of adapting to well-known environmental conditions. Therefore, molecularly imprinted hydrogels play an important role in the development of modern drug delivery systems.

## 1. Introduction

Molecular Imprinted Polymers (MIPs) are polymeric systems that possess a unique property to recognize a specific molecule or group of structurally related molecules. MIPs selective recognition’s property is determined during the preparation of polymer using a template molecule together with appropriate monomers in a solvent. MIPs are prepared in the presence of template molecules that can be subsequently removed, which determine MIP’s cavity selectivity for a specific template or compounds structurally related to this template [1,2]. Created tailor-made sites gain the property to selectively recognize the template molecule’s size, shape, and functional groups.

The first reported molecular imprinting concept was proposed in 1931 by Polyakov [3] as “unusual adsorption properties of silica particles prepared using a novel synthesis procedure”. The mentioned “unusual adsorption properties” have been reported using numerous polymers, which have been subsequently named as molecularly imprinted polymers—MIPs.

There are three methods to form molecular imprinting. The first one is a covalent method based on reversible covalent bonds, introduced by Wulff in 1995 [4], the second is a method proposed by Mosbach in 1994 [5], which is based on non-covalent interactions between templates-imprinted molecules- and functional monomers and the last one is semi-covalent, reported by Whitcombe et al. [6], in which subsequent rebinding by non-covalent bond can be created after a covalently bounded template is removed (Figure 1) [6,7]. 

The covalent imprinting method is based on creating a covalent bond between the template and the appropriate monomer. Then, during polymerization, the covalent linkage is cleaved, and subsequently, the template is removed from the MIP matrix. Rebinding of the previously removed template causes reappearance of the same covalent linkage. Since the formation of identical rebinding linkages requires rapidly reversible covalent interactions between templates and appropriate monomers, the number of suitable templates for covalent imprinting is limited. Additionally, the robust nature of the covalent interactions and consequent slow dissociation and binding makes it hard to reach thermodynamic equilibrium. The second method, non-covalent imprinting, has no such restrictions and is the most frequently used due to its simplicity [8]. By using an appropriate solvent, the formed various interactions such as hydrogen bonds, π-π and ionic interactions, van der Waals forces, etc., generate template-monomer complexes. After removing the template from the MIP matrix, the interactions can be easily recreated. The removed template can be rebound via the same non-covalent interactions as before polymerization. Therefore, the range of applicative compounds which can be imprinted via non-covalent imprinting is expanded, and non-covalent imprinting has become the most popular and general synthesis strategy for Molecular Imprinting Technology (MIT). The third type of imprinting method is semi-covalent, defined as subsequent rebinding by non-covalent bond after a covalently bounded template is removed. This semi-covalent approach was firstly reported by Whitcombe et al. [6] and offered an intermediate alternative in which the template is bound covalently to functional monomer since the template rebinding is based on non-covalent interactions. Semi-covalent bond can be characterized by the high affinity of covalent binding and mild operation conditions of non-covalent rebinding. The schematic diagram of non-covalent imprinting mechanisms is presented in Figure 2 [7].

In comparison to other well-known recognition systems, MIPs have received considerable attention. Thanks to that, MIPs are widely used in various fields such as purification [9], separation [10], and catalysis [11], and degradation processes [12] but also they have become attractive in drug delivery [13], artificial antibodies [14], or biosensing [15]. The widespread use of MIPs is an aftermath of their favorable characteristics, such as high physical stability to harsh chemical and physical conditions, straightforward preparation, remarkable robustness, excellent reusability, and low-cost synthesis [7,16]. Whereas MIPs present a wide range of advantages, there is some drawback that should be considered. One of them is the design of a new MIP system that will be suitable for a specific template molecule usually requires a lot of work and time to estimate the best synthesis conditions that allow obtaining the intended material. Before finding the optimum conditions, there is a necessity to continually change various experimental parameters [13].

## 2. Fundamentals of MIPs

Molecularly imprinted polymers are polymeric matrices that are moulds for the formation of template complementary binding areas. They can be programmed to recognize a large variety of structures with antibody-like affinities and selectivities. In addition to the already mentioned advantages, these properties have made MIPs attractive in various fields. The main applications of molecularly imprinted polymers are presented in Figure 3 [17].

### 2.1. Essential Elements of Molecular Imprinting

Generally, MIPs are synthesized using a functional monomer, template, cross-linker, a polymerization initiator, and an appropriate solvent. In short, MIPs are prepared by mixing the mentioned molecules, and then, this pre-polymerization mixture has to be irradiated with UV light or subjected to heat to initiate polymerization [8]. As polymerization is affected by many factors, MIPs can be modified by the appropriate choice of synthesis conditions. There is a possibility to obtain various MIPs with specific properties due to changes of factors such as type and amount of monomer, initiator, cross-linker, and solvent. Additionally, the time and temperature of polymerization reaction also play an important role in creating MIPs with superior targeted properties [16].

The central importance of MIP structure is a template, which can direct the organization of the functional group’s pendant to the functional monomers in the molecular imprinting process. Templates should be inert under the polymerization conditions during free radical polymerization [8]. The main goal of molecularly imprinted technology is to create MIPs compared with biological receptors in specificity. Thanks to that, MIPs might replace those entities in real-life applications. Since there is a lot of requirements that should be met, the three mentioned ahead make the template an ideal candidate—it should exhibit excellent chemical stability during the polymerization reaction, it should contain functional groups that do not prevent polymerization, and it should contain functional groups that can form complexes with functional monomers [6]. Additionally, established imprinted small organic molecules such as pharmaceuticals, pesticides, or amino acids are well-known and commonly used. Furthermore, a lot of research proves that not only small molecules are suitable for Molecular Imprinting Technology. Since small molecules have a lot of advantages—like being more rigid to form well-defined binding cavities during the imprinting process—there are some of the protocols that reported using larger organic entities like proteins or even cells. As only a few protocols are reported, imprinting larger organic compounds containing secondary or tertiary structures is still a challenge because these structures may be affected when exposed to the thermal or photolytic treatment involved in the synthesis of MIPs. The rebinding process is also more complicated when using such individuals as large templates, as they do not penetrate the polymeric network easily to reoccupy the binding cavities [8].

It is essential to select a suitable functional monomer that can strongly and selectively react with the template to form specific complexes. Generally, the functional monomers are responsible for the binding interactions present in the imprinted binding sites during the imprinting process. As reported in many protocols, for non-covalent molecular imprinting reaction, the monomer is used in excess of the number of moles of the template to form template-functional monomers assemblies. To maximize expected complex formation, matching the template’s functionality with the monomer’s functionality plays a crucial role. The imprinting effects increase when the template’s functionality is matched with the functionality of the monomer in a complementary fashion, like an H-bond donor with an H-bond acceptor [7]. The amount of monomers that can be used in molecular imprinting is limited. It is imperative to synthesize new functional monomers that form specific interactions with the templates. Typically, monomers include two independent types of units—the recognition unit and the unit, which can be polymerized [16]. Figure 4 presents widely used functional monomers [8].

The amount of cross-linker used in the polymerization process also plays an important role in MIPs properties. Too low cross-linker causes unstable mechanical properties, whereas too high amount will reduce the number of recognition areas per unit mass of MIPs. The primary role of a cross-linker is to form a highly cross-linked polymer. Cross-linker is involved in fixing monomers around template molecules, thus forming a cross-linked polymer. The main aim is to develop a highly cross-linked polymer even after removing templates. Generally, the amount and type of used cross-linker regulate the selectivity and binding capacity of MIPs [8,16]. In MIPs structure, cross-linker fulfills three major functions: controlling the morphology of the polymer matrix, serving to stabilize the imprinted binding site, and giving mechanical stability to the polymer matrix. The structures of commonly used cross-linking agents in molecular imprinting techniques are presented in Figure 5 [8].

The most useful reactions for preparing MIPs are free radical polymerization (FRP), photopolymerization, and electropolymerization [16]. Plenty of initiators with different chemical properties can be used as the radical source in a free radical polymerization (Figure 6) [8]. In comparison to the monomers, initiators are used at low levels, e.g., 1 wt.%, or 1 mol.% with respect to the total amount of moles of polymerizable double bonds. Moreover, there are several existing ways in which the rate control and mode of decomposition of an initiator to radicals, including heat, light, and by chemical/electrochemical means, depending upon its chemical nature, can be achieved [8].

The last chemical individuum that strongly impacts proper MIPs formation is solvent. During polymerization, it generally plays an important role as dispersion media and pore-forming agent. Commonly used solvents include 2-methoxyethanol, methanol, tetrahydrofuran, acetonitrile, dichloromethane, chloroform, N,N-dimethylformamide, and toluene [18]. Porogenic solvent needs to have the following features: all of the used chemical individuals should be well-soluble in the chosen solvent, the solvent should produce large pores to assure flow-through properties of the polymer, and the last on, the solvent should possess low polarity to avoid interferences during complex formation between imprinted molecules and monomers, which is important to obtain high selectivity MIPs [8]. The interactions between templates and monomers depend on the used solvent’s polarity. Non-polar or low polar solvents such as chloroform are used for non-covalent imprinting. Due to that, the obtained MIPs gain good imprinting efficiency because the adsorption properties and morphology of polymer depend on the type of used solvent [16].

### 2.2. Molecular Imprinting in Drug Delivery

Molecular imprinting is one of the most promising ways to recreate biological molecular recognition and mimic properties of antibodies and enzymes in synthetic materials. Many researchers are trying to mimic molecular interactions present in these systems as high recognition characteristics seem to be the fundamental requirement of living systems. Therefore, most approaches focus on creating a binding cavity in which functional chemical groups may be strictly positioned. The mechanism of selective recognition and subsequent drug release from the MIPs structure is presented in Figure 7.

Whereas MIPs are widely used in various areas of science, the area with one of the greatest potential seems to be pharmaceutical chemistry, where MIPs, as synthetic molecularly selective receptors, may be used in therapeutics or medical therapy [17]. Due to properties such as biocompatibility, low toxicity, and biodegradability, MIPs receive extensive attention as drug delivery systems (DDS). MIPs have already been used as selective oral adsorbents for cholesterol [19] and imprinted bile acid sequestrants [20,21]. Commonly, MIPs are widely used as DDS in various diseases such as cancer [22], arrhythmia [23,24], avitaminosis [25], cardiovascular and cerebrovascular disease [26], inflammation [27], addictive disease [28,29] and other [30]. There are also several applications in which MIPs are incorporated into membranes to be used for bio-separation and bio-purification [31]. MIPs are also used in controlled release delivery systems. It is reported that MIPs are widely used for modifying drug release from solid dosage forms, which results in tuned composition release. Whereas many studies are based on a simple modification of non-imprinted polymers, there is also a huge potential in another MIPs application area—intelligent drug release. This release refers to the predictable release of the therapeutic agent in response to specific stimuli like the presence of another specific molecule or change in pH. An example of this intelligent release might be a cell surface epitope. The general mechanism of drug release from the cell surface is illustrated in Figure 8 [17].

## 3. MIP Challenges in Transdermal Delivery

The oral route is the most common and convenient route of drug administration. Low oral bioavailability and side effects restricted oral administration of most drugs with poor solubility in clinical use [30]. However, there are a lot of medicines that implementation by the oral route is characterized by a low adsorption rate from the gastrointestinal tract or extensive first-pass metabolism [32,33]. In these examples, the skin may be an alternative route for applying drugs, as it has been widely used as a route of administration for local and systemic drugs [34,35]. Transport across the biological barrier creates a problem resulting from the interaction or reaction of the experimental drug with the epithelial barrier [36]. The use of MIT may be integrated to design DDS with properties tuned with pharmaceutical agents produced for dermal or transdermal use [2].

There are a lot of studies that involve MIPs in transdermal drug delivery applications. In one of them, a molecularly imprinted nicotine transdermal system was proposed. In this study reported in 2014 by Ruela et al. [28], an imprinted matrix was prepared by free radical polymerization of a copolymer of methacrylic acid and ethylene glycol dimethacrylate using the bulk technique in the presence of nicotine, which acted as the template. The bulk material was then crushed to obtain particles of 75–106 µm. These particles were dispersed in mineral oil or propylene glycol and formed into a disk with a surface equal to 1.8 cm^2^. As results showed, mineral oil was the most promising vehicle due to its hydrophobic characteristics, which improve the molecular recognition of nicotine in MIP particles. Polymeric particles present in the transdermal system differed in polarity. In this study, the pH of the nicotine imprinted polymeric delivery system was similar to the skin’s pH. The imprinted polymers were characterized using various techniques to study the morphology of the particles, drug-polymer interactions, and compatibility. The results of controlled release were compared with the commercially available product—Nicopath^®^. Results obtained after in vitro experiments showed that the amounts of permeated nicotine from the imprinted matrix were similar to commercial patches. The results are 655 and 709 µg cm^−2^ for 24 h, respectively. According to the results obtained in the study made by Ruela, a MIP created with a nicotine template showed promising results. It was demonstrated that non-covalent MIP drug interactions might modify the profile of drug release and skin penetration. Additional studies, such as FT-IR or SEM, also confirmed that prepared MIPs with nicotine as a template have high thermal stability and are resistant to chemical degradation. Although both molecularly imprinted polymer and non-molecular imprinted polymer could bind the templates in their matrixes, MIPs showed better performance during the transdermal release. It is caused by the presence of selective recognition sites in the MIP structure [28,34].

In the further studies made by the same research group, experiments involving the synthesis of several MIPs using precipitation polymerization technique to find optimized materials able to selectively absorb nicotine were performed. As a result, release and skin permeation by nicotine was optimized using MIPs synthesized by precipitation polymerization technique. Obtained polymers showed improved adsorption capacity and selectivity, additionally, MIPs were also able to modulate the transdermal delivery of templated nicotine [29].

Another application of imprinted polymers is shown in one of the studies presented by Bodhibukkana et al. [37]. MIPs were used as a composite material integrated with cellulose to form a membrane to improve the biocompatibility of the transdermal system (Figure 9). As previous research shows, the cellulose membrane is a biocompatible and biodegradable material with good mechanical properties.

These studies aimed to modify the cellulose membrane with a thin layer of *R*-propranolol or *S*-propranolol entrapped in MIPs structure. The results showed the potential of molecularly imprinted polymer composite membranes based on cellulose in controlling *S*-propranolol release into the skin. The degree of stereoselectivity demonstrated a higher therapeutic advantage when considering the two enantiomers of propranolol (*R*/*S*). Due to selectivity towards *S*-propranolol of the MIPs present in the surface of the cellulose membranes, a limited release was achieved [34,37].

## 4. Basic Characteristics of Hydrogels

Polymeric hydrogel networks are insoluble, cross-linked, and composed of hydrophilic homo- or hetero-co-polymers, absorbing significant amounts of water and retaining their shape without dissolving. Cross-links within hydrogels may be covalent bonds, permanent entanglements, ionic interactions, or microcrystalline regions incorporating various chains. Loading therapeutics into the hydrogel network takes place using one of two possible scenarios. One of them refers to producing the appropriate gel in the presence of the drug, whereas the second is a path of firstly synthesizing the gel and then loading therapeutic into the gel [38]. Using appropriate monomers with defined properties allows the formed hydrogels to be environmentally responsive, for example, toward changes in pH. Generally, molecular imprinting technologies allow hydrogels to: recognize and selectively bind the specific substrate into the hydrogel. Hydrogels have many advantages compared with other imprinting materials, such as milder synthesis conditions at lower temperatures or relatively high solubility of biological templates like DNA, protein, nucleic acid, etc. Due to that, molecularly imprinted hydrogels play an important role in modern drug delivery systems [39].

The most valuable feature of hydrogels in drug delivery systems is their ability to control diffusion and ability to amplify the microscopic events, which occur at the mesh chain level into macroscopic phenomena [38,40,41]. It is well known that the delivery of certain drugs directly to localized sites beneath the skin is highly desirable in some cases since it would allow local pathology to be treated without significant systemic side effects [42]. Additionally, there are some benefits of transdermal drug delivery within using hydrogels. The drugs dosage can be interrupted on-demand by simply removing the devices, and that drugs can bypass hepatic first-pass metabolism. Using hydrogels is also beneficial because of their dual structure, involving a macroscale three-dimensional macromolecular network with a highly hydrated microscale environment where the former characteristic supplies necessary macroscale rigidity, whereas the latter provides the potential for relatively efficient mass transfer [43]. What is important, swollen hydrogels, due to their high water content, may provide a better feeling for the skin in comparison to conventional ointment and patches [40].

The behavior of hydrogels in a changing environment is presented in Figure 10. Immersing a dry hydrogel in a compatible solvent causes the solvent movement into the hydrogel polymer chain followed by volume expansion and macromolecular rearrangement depending on the extent of crosslinking within the network (presented in Figure 10). Two factors are decisive for the rate at which a polymer expands or swells—the rates of polymer-chain relaxation and solvent penetration into the hydrogel network. Moving from an unperturbed, glassy state to a solvated, rubbery hydrogel state leads to unlimited exchange in transport. This is an important feature for swelling-controlled hydrogels, in which we can obtain a zero-order release or constant release rate. These release rates can be achieved by keeping the constant rate of solvent front penetration, which should be smaller than the drug diffusion.

In Figure 10, a modified hydrogel molecule is shown. The presented hydrogel molecules may contain a specific chemical/biological species along their backbone chain to obtain sensitivity to environmental hydrogels. This feature may be achieved by controlling drug transport by swelling controlled systems (i.e., drug-loaded dry state with water uptake) or swellable systems (e.g., pH, temperature, etc.). In a widely used Fickian model of release kinetics, the relaxation rate is high, resulting in the rate-limiting diffusion process. Thus, the release rate of the drug is proportional to the concentration gradient between the drug source and the environment. The achieved rate is proportional to the concentration gradient between the drug source and the surroundings. The main aim is to find a drug source to achieve zero-order release. Many strategies try to achieve zero-order release, such as biodegradable systems with solvent penetration moving with similar velocities the outer eroding [38,44].

One of the new methods is to obtain hydrogel with macromolecular memory for the drug within the network and delay the transport of drug from the hydrogel matrix by the presence of interactions with various functional groups within the network. This can be achieved by using molecular imprinting methods presented in Figure 10 Interactions between the drug and matrix cavities slow drug release from the hydrogel. This type of hydrogel optimization of slowed release, caused by the amount and strength of functional monomer interactions, crosslinking structure, and mobility of polymer chains, might be a potentially synthetic solid way to gain many hydrogels [38].

## 5. Mechanism of Controlled Release within Molecular Imprinted Hydrogels

Molecular Imprinted Hydrogels can be classified as anionic, cationic, or neutral, which also determines their behavior. Thermodynamically, the swelling behavior of the hydrogels network is related to the balance between the polymer-water Gibbs free energy of mixing and the Gibbs free energy associated with the elastic nature of the entire polymer [45]. The quantities of the mentioned free energies become equal when achieving the swelling equilibrium [46]. What highlights the hydrogels from others is the advantage of milder synthesis conditions at lower temperatures and in aqueous mediums regarding the fragility and solubility of biological templates, including DNA, protein, or even nucleic acids [39]. Two main solvent-activated systems can be indicated—an osmotic-controlled system and a swelling-controlled system—the rate of water influx controls the overall rate of the drug release. The controlled drug release mechanism is based on water diffusion and polymer chain relaxation [46]. Generally, the time dependence of the drug release rate can be determined by the rate of water diffusion and chain relaxation [47].

It is well-known that the limitations of transdermal drug delivery are controlled by skin anatomy. Generally, the skin permits a painless and compliant network for systemic drug administration [48]. The fact that the skin has evolved and thus impedes the flux of toxins into the body and minimizes water loss means that it naturally has a low permeability to the penetration of foreign molecules. Because the skin provides a barrier to the delivery of many drugs, various chemical additives have been tested to achieve better results in transdermal penetration. Chemical penetration additives offer many advantages, such as design flexibility with formulation chemistry and a more accessible patch application over a large area [49]. The mentioned transdermal patches have been widely helpful in developing new applications for existing therapeutics and reducing first-pass drug-degradation effects. Patches also gain the ability to reduce some side effects. For example, estradiol patches are commonly used and, in contrast to the popular oral formulations, do not cause liver damage [50].

Whereas the mechanism of controlled release of the drug from hydrogel structure is relatively easy to design, implementing imprinted recognition release systems requires consideration of many environmental impacts and the expected properties of the desired hydrogel. Basically, the controlled release mechanism and associated swelling characteristics of polyhydrogels’ networks result from cross-links (also known as tie-points or junctions), permanent entanglements, ionic interactions, or microcrystalline regions incorporating various chains. In general, as an analyte replaces pendant analyte groups (attached to the copolymer chains), the polymeric network loses effective cross-links, opening the network’s mesh size and regulating the release. Otherwise, as an analyte decreases in concentration within the bulk phase, the molecule rebinds with the analyte groups attached to the copolymer chains, which role is to close the network structure [46].

It is not a surprise that one of the most desirable controlled drug delivery applications is stimuli-responsive hydrogels that can modulate the release in response to pH, temperature, ionic strength, electric field, or specific analyte concentration differences. The most important feature of these systems is that they can be designed to operate within a particular human body area due to the possibility of adapting to well-known environmental conditions [46,47].

### 5.1. Stimuli-Responsive MIP Hydrogels

The need for creating intelligent materials based on chemical compounds that can mimic the natural receptors inspires the development of imprinting technologies and expand the MIPs synthesis into the synthesis of stimuli-responsive MIPs (SR-MIPs) by stimuli-responsive technology for molecular imprinting. SR-MIPs included thermo-responsive MIPs, pH-responsive MIPs, dual- or multiple-responsive MIPs, and other-responsive MIPs. Due to their great applications properties, these intelligent polymers play an important role in many fields such as drug delivery, biotechnology, separation science, cell encapsulation in biochemistry, and chemo-biosensing [51]. The combination of stimuli-responsivity and Molecular Imprinting Technology helps to obtain valuable functionalities. Generally, imprinting provides a high loading capacity of specific molecules, whereas the ability to respond to stimuli modulates the affinity to network for the target molecules. The whole process provides a regulatory or switching capability of the release process [52].

Connecting molecularly imprinting technology with the synthesis of stimuli-responsive hydrogels requires conducting the polymerization reaction in the presence of a template in the conformation corresponding to the minimum energy. Imprinted cavities’ recognition properties after swelling can be maintained only if the network folds back into the conformation adopted during the synthesis [53]. Generally, when the centers of molecular recognition are present in the stimuli-responsive hydrogel, the conformation of the receptors may be deformed or re-constituted as a function of an external or a physiological signal. There are plenty of functions that stimuli-responsive polymeric hydrogels can perform, such as selectively and effectively load of a particular drug, releasing the drug at a rate modulated by a stimulus, and uptake the released drug again from the environment if the drug remains around the hydrogel when the stimulus stop or diminishes its intensity and the cavities are reformed (Figure 11) [52].

Generalizing, stimuli-responsive imprinted hydrogels can be synthesized by combining responsive monomers with functional monomers that interact with the appropriate drug molecules. After polymerization reaction, during hydrogel swelling, the structure of receptors is altered, and the drug is released. The receptors can be reconstituted following stimulus disappear or decrease in its intensity. As a consequence, the release slows down or even stops. Whereas there is a necessity to recognize cavities structure after several swelling/collapse cycles, optimizing stimuli-responsive imprinted hydrogels is still challenging [52,54].

#### 5.1.1. Thermo-Responsive Hydrogels

The thermo-responsive gels have been widely used as smart materials in various fields such as drug delivery systems, tissue engineering, or even cell encapsulation in biochemistry [55,56,57,58,59]. Thermo-responsive MIPs have gained the researchers’ curiosity due to the similar recognition mechanisms to the proteins from natural systems and the ability of the hydrogels to swell or deswell thanks to changes in temperature in the surrounding area [40]. Therefore, thermo-responsive polymeric hydrogels can be used in the design of protein-imprinted polymeric materials, which have already been reported in many applications [55,56,59,60]. The important thing is that there is dependence on the availability of binding sites from the MIP structure based on cross-linking. Highly cross-linked MIPs have a more rigid structure thus, the number of binding sites is limited, whereas lightly cross-linked polymer gels can undergo reversible swelling and shrink in response to environmental temperature changes [16]. There are two classes of thermo-responsive hydrogel materials—positive and negative temperature-responsive systems. The main difference is critical solution temperature—the positive temperature-responsive hydrogels have an upper critical solution temperature (UCST), so it means that they contract upon cooling below the UCST. In contrast, the negative-sensitive hydrogels have a lower critical solution temperature (LCST), and they contract upon heating above the mentioned LCST [40]. Generally, thermo-responsive polymers contain both hydrophilic and hydrophobic groups. Due to that, they can form appropriate structures, swelling and shrinking, in response to temperature changes. The mechanism of this response is based on hydrogen bond interactions. It is well known that in lower temperatures, the hydrogen bond interactions are formed between hydrophilic areas in polymer chains and templates, whereas in higher temperatures, higher than the low-critical solution temperature (LCST), the hydrogen bond interactions are destroyed, thus hydrophobic interactions increase. An increase of hydrophobic bond interactions causes the aggregation of polymer chains and then contraction of the gel network [51].

The most known thermo- responsive polymer is poly(N-isopropylacrylamide) (PNIPAAm). Its low-critical solution temperature is around 32 °C in an aqueous solution, so it means that due to the close to natural body temperature, it may be used widely in smart drug delivery systems [61,62]. Generally, the combination of thermo-responsive properties with Molecular Imprinting Technology can develop networks that provide a promising synthetic strategy ensuring the system responds rapidly to external temperature changes. The schematic mechanism of thermo-responsive hydrogel’s action is presented in Figure 12. It is shown that the template can be easily removed from the MIPs network by reducing the external temperature (Figure 12) [51].

A lot of studies are reported, showing that the N-isopropylacrylamide (NIPAAm) can be used as a functional monomer for preparing thermo-responsive MIPs, applicable in various fields. The NIPAAm has been used for many target species such as proteins [59,63,64,65], organic molecules (like 4-aminopyridine) [63], cisplatin [66], or even metal ions (like Cu^2+^ ions) [67].

Wang et al. [68] reported the results of research in which a preparation of pH/thermo-responsive MIPs by frontal polymerization using acrylic acid and N-isopropylacrylamide (NIPAAm) was performed. The proposed MIPs were applied to deliver Gemifloxacin, a fourth-generation fluoroquinolone antibiotic that acts by inhibiting DNA gyrase and topoisomerase IV. The reported data showed that the obtained drug delivery devices based on MIPs possessed higher relative bioavailability of Gemifloxacin than those of the corresponding non-imprinted polymers [68].

One of the recently studied MIP hydrogels with the thermo-responsive feature is a molecularly imprinted polymer based on konjac glucomannan (polysaccharide) imprinted with 5-fluorouracil as a template reported by Ann et al. [69]. 5-Fluorouracil is a compound with a high affinity to a range of tumors such as gastric, intestinal, pancreatic, ovarian, liver, brain, breast, etc. In the reported studies, a novel thermo-responsive MIP was prepared by graft copolymerization using konjac glucomannan (KGM) as a matrix, N-isopropylacrylamide (NIPAAm) as a thermo-responsive monomer, acrylamide (AM) as co-monomer, N,N’-methylenebis(acrylamide) (NBAM) as a cross-linking agent, and 5-fluorouracil (5-Fu) as a template (Figure 13).

5-Fluorouracil selective MIP was characterized by thermo-responsive features. The results showed that the system could quickly respond to an external change in temperature. The swelling or shrinking of the imprinted sites resulted in the adsorption or desorption of 5-fluorouracil. As a result, the prepared MIPs could be used as a sustained-release network controlling the release of 5-Fu by changing the environmental temperature. Obtained data of the release kinetics was fit with the Higuchi release model [69].

#### 5.1.2. pH-Responsive Hydrogels

Hydrogels, which are pH-responsive, must contain many chemical groups that can be easily ionized. Carboxyl or amino groups can be noted as examples of these groups that can, accordingly, donate or accept a proton, which determines the pH-sensitivity feature of the entire MIPs. As in the case of thermo-sensitivity, the mechanism of pH-response is based on hydrogen bonds interaction between the chains and template [51]. When the chemical group is ionized during changes of environmental pH, at the same time, the hydrogen bonds between chains are destroyed, which causes a decrease in the crosslinking points in the hydrogel network. This results in a discontinuous change in the hydrogel volume [70]. Considering the possibilities of ionized groups, pH-responsive polymers can be divided into two types—anionic and cationic ones. Within the anionic types, the most useful group is a carboxyl group, which can be protonated and thus determine hydrophobic interactions at low pH. Additionally, a low pH environment cause leading the volume shrinkage. Opposite to that, at high pH, the behavior of hydrogel is quite different. In these conditions, carboxyl groups dissociate into carboxylate ions, resulting in a high charge density in the polymer network, which causes swelling. Similarly, the pH-responsive feature of the cationic hydrogel network is dependent on the protonation of basic groups in the polymer chains (e.g., amino groups or pyridine groups). At low pH, basic cationic groups are protonated, which leads to internal charge repulsions between neighboring protonated groups. In contrast, at higher pH, the groups become less ionized, resulting in a reduction in the overall hydrodynamic diameter of the polymer [51].

The first reported pH-responsive MIPs were proposed by Tao et al. [71]. In this research, novel pH-responsive MIPs by using amylose as the host matrix, bisphenol A (BPA) as the template, and acrylic acid (AA) as the co-functional monomer prepared. Changing the acidity of the entire solution could reversibly control the rebinding ability towards the template. In that case, the rebinding ability of polymers decreased with the increasing pH of the solution. Comparing two pH conditions, pH_1_ = 4.5 and pH_2_ = 8.5, the binding amount was, accordingly, 2.5 µM·g^−1^ and 1.0 µM·g^−1^ [51,71]. The higher pH caused the loss of the MIP affinity for bisphenol A because of the conformational changes in the amylose chains caused by the electrostatic repulsions among the ionized groups of acrylic acid and the subsequent disruption of the imprinted cavities (Figure 14) [52].

The example of pH-responsive MIP prepared for controlled release of dexamethasone-21-phosphate disodium salt (DXP) is nanosphere/hydrogel composite reported by Wang et al. [72]. The chosen DXP is a potential coating for implantable biosensors that should improve their biocompatibility [51]. As results show, the DXP release rate from the MIP structure increased significantly with a decrease of pH value, while the DXP release rate from the non-imprinted polymer structure didn’t change with the pH change (Figure 14).

The behavior of DXP in various pH conditions can be explained by interactions between the template and polymer network. When the pH value decreases, some template anions are protonated, which causes weak ionic interactions. It subsequently leads to a faster drug release rate [72]. Obtained hydrogel has been applied in glucose sensors to improve their biocompatibility as well as their lifetime. The results showed that the obtained DXP hydrogel MIPs can potentially suppress the inflammation response, causing an increase in the pH of the imprinted sensors, effectively improving their lifespan [51].

Interesting studies of dual drug release with materials based on poly(L-lactide)-*co*-polyethylene glycol-*co*-poly(L-lactide) dimethacrylate as a degradable polymeric cross-linker were reported by Xu et al. in 2016 [73]. They used acrylic acid and N-isopropylacrylamide as monomers, anti-cancer drug DOX, and the antibiotic tetracycline, which both were used as templates loaded into the hydrogels with dual drug loading efficiency. The drug release at pH 7.4 and 1.2 were examined. The obtained results confirmed the original thesis and showed that the synthesized copolymers are pH-responsive, shrinking at pH = 1.3 and swelling at pH = 7.4. As experiments demonstrated, the dual-drug-loaded hydrogels released drugs in different patterns and successfully killed targeted cells [73].

#### 5.1.3. Dual/Multiple-Responsive MIPs

The dual/multiple-responsive hydrogels are polymeric systems responsive to two or more external stimuli [51]. While the studies on dual stimuli-responsive polymers are widely common, they are relatively less explored than single-responsive polymers. However, dual-responsive MIPs gain the researcher’s curiosity due to their feature, such as enhancing the versatility of polymeric materials as they allow tuning of their properties in multiple ways rather than in single-responsive polymers [16]. From the synthetic point of view, by using suitable monomers, the dual-responsive MIPs may be obtained by transforming the single-responsive polymer by replacing the traditional functional monomer with a stimuli-responsive functional monomer [51]. Generally, dual-responsive polymers mainly include: magnetic/photo; magnetic/thermo; thermo/pH; thermo/photo; and thermo/salt dual responsive MIPs. The mentioned types of well-reported dual-responsive MIPs useful in various fields are presented in Table 1.

One of the most interesting uses of dual-responsive MIPs is reported by Zhao et al. [77], a multi-responsive MIP consisting of thermo-responsive and salt-responsive MIPs hydrogel. The research presents dual-responsive MIPs hydrogel for BSA by self-assembly of a basic functional monomer N-[3-(dimethylamino)propyl]methacrylamide (DMAPMA) with bovine serum albumin (BSA) that can be polymerized in the presence of NIPAAm. Obtained dual-responsive polymeric hydrogel proved that it possesses a clear memory of the template protein and can respond to changes in temperature and ionic strength. In the recognition process mechanism, salt ions play an important role in screening the electrostatic interactions between protein molecules and the charged polymer chains. It was observed that the increase of salt concentration caused a screening of electrostatic interactions between polymer chains, while the addition of NaCl to the adjusted volume of polymer caused inhibition. The demonstrated features of obtained dual-responsive MIPs made them attractive for applications such as solid electrolyte membranes, electrode devices, protein delivery agents, and sensors with the controlled release [80].

There are also some studies reporting multiple-responsive MIPs, however, the amount of such systems is much lower than dual-response MIPs. One example proposed by Chen et al. [78] is multi-responsive protein imprinted polymers responsive to temperature, the corresponding template protein, and salt concentration, which results in specific volume shrinking. Cytochrome c or lysozyme were used as templates, NIPAAm as a major functional monomer, MAA and AAm as functional co-monomers, and N,N-methylenebisacrylamide as a cross-linker. As the results showed, the combination of molecular imprinting technique and a stimuli-responsive feature may be useful for preparing protein-responsive polymeric hydrogels that can undergo specific binding and shrinking in the presence of template [65].

Recently, interesting research was reported by Wang et al. [81] based on the fabrication of core-shell imprinted nanospheres with multiple responsive properties. The scheme of the main synthesis is shown in Figure 15.

As results showed, fabricated magnetic core-shell MIP nanospheres, with an imprinted layer with a thickness ranging from 40 to 150 nm, gain fine hydrophilicity, high binding capacity, and favorable selectivity adsorption in aqueous solution. Obtained polymers were also used in the application of drug delivery systems and expressed a sustained release effect triggered by temperature and UV light [81].

#### 5.1.4. Other-Responsive Hydrogels

There is a lot of possibilities to obtain a stimuli-responsive polymeric hydrogel. Hydrogels may be fabricated to be responsive for various, additional to mentioned in previous paragraphs, external stimuli such as light, magnetic field, or other stimuli including salt ions and biomolecules [16]. Whereas many studies are reporting all of the mentioned external stimuli responsive materials, from pharmaceutical point of view, the most interesting thought is to be the last, using biomolecule responsive polymeric networks. Recently, biomolecule-responsive hydrogels have become more important for drug delivery systems and molecular diagnostics because they can sense the target biomolecule, which results in structural changes. The mechanism of biomolecule-responsive MIPs is based on changes in the volume as a function of the concentration of a target biomolecule such as carbohydrates or proteins [82]. The fabricated MIP’s hydrogels may be an optimal way to overcome tumors as some studies confirmed that suppositions. A good example is a research presented by Miyata, who prepared tumor-marker-imprinted hydrogel, which can shrink in the presence of a target-tumor marker glycoprotein by using various cross-linkers (low-molecular-weight/high-molecular-weight). The obtained results provided a basis for developing useful biomolecule-responsive hydrogels and permitted the knowledge of critical factors of molecular imprinting. The presented studies were focused not only on the proper preparation of biomolecule-responsive hydrogel but also on the effect of the molecular weight of cross-linkers on the glycoprotein-responsive behavior of imprinted hydrogels. Generally, whereas long chains of the high-molecular-weight cross-linker and network chains undergo conformational changes by complex formation of ligands with a target glycoprotein, short chains of the low-molecular-weight cross-linkers do not undergo these changes [83].

## 6. Conclusions and Future Work

Molecularly imprinted polymers are promising materials in the synthesis of advanced drug delivery networks due to their ability to increase release times and extend the residency of the entire drug. As the presented studies showed, the main application fields in which MIPs hydrogels can be used are sustained release, controlled release, and targeted delivery system based on its distinct advantages. The future perspectives for transdermal imprinted drug delivery devices are very promising because of noted enormous progress in synthetic and material approaches. The main advantage of using MIPs is their high stability, from which there is a possibility to note: resistance to pressure, high temperatures, extreme pH, and possibility for long-term storage.

In this review, there was also summarized a mechanism and application of various stimuli-responsive MIPs. As presented, stimuli-responsive MIPs can be divided into single-responsive and multi-responsive MIPs. The main conclusion is that the second group has received more attention between non-stimuli and stimuli-responsive MIPs because of their excellent properties. One of the main benefits is their response to external stimuli, which makes it possible to alter their volume and affinity for target molecules by changing the environmental conditions.

Although various achievements have been attained in molecularly imprinted technology and stimuli-responsive MIPs, there are still lots of development challenges and opportunities. For instance, it is still challenging to transfer the imprinting process from organic to aqueous phase, reaching the level of natural molecular recognition, exploring various stimuli-responsive systems to develop stimuli-responsive MIPs, and developing within dual/multiple-responsive MIPs with good biocompatibility with increasing requirements for functional polymer materials.

## Figures and Tables

**Figure 1 polymers-14-00640-f001:**
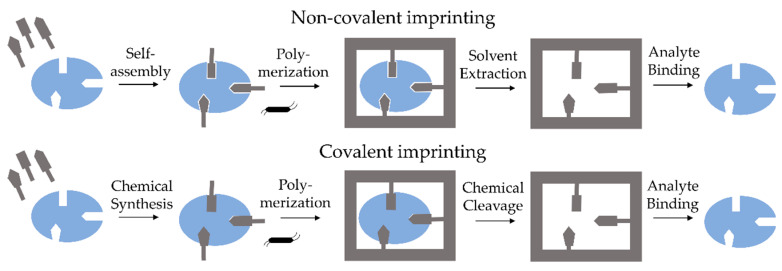
Schematic representation of covalent and non-covalent mechanisms of molecularly imprinted procedures.

**Figure 2 polymers-14-00640-f002:**
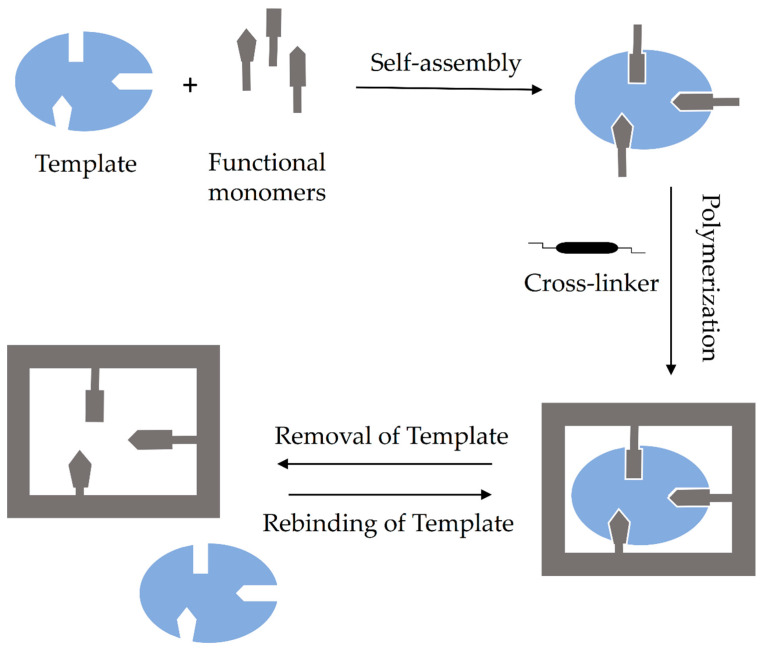
Schematic diagram of the non-covalent method for molecular imprinting process.

**Figure 3 polymers-14-00640-f003:**
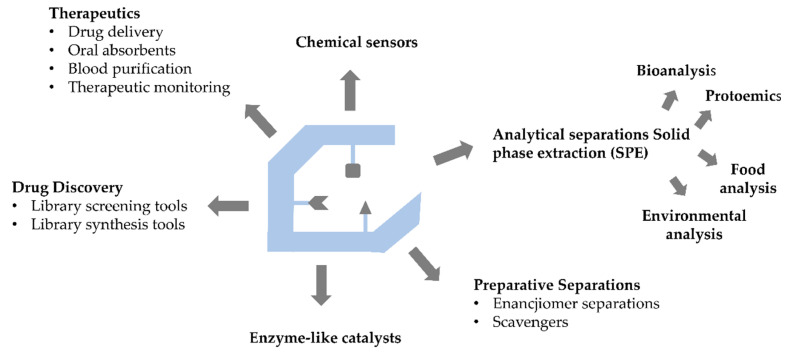
Scheme of the main applications for MIPs.

**Figure 4 polymers-14-00640-f004:**
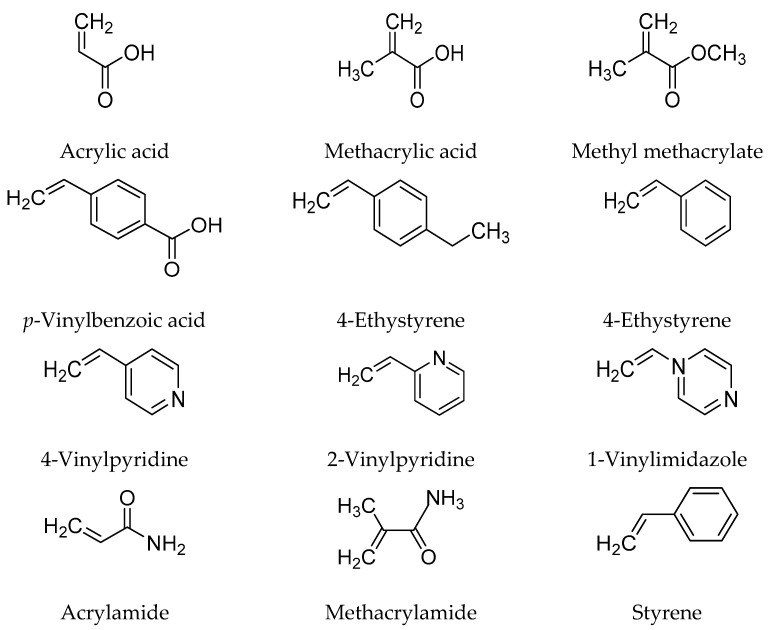
Scheme of commonly used monomers for non-covalent molecularly imprinted technique.

**Figure 5 polymers-14-00640-f005:**
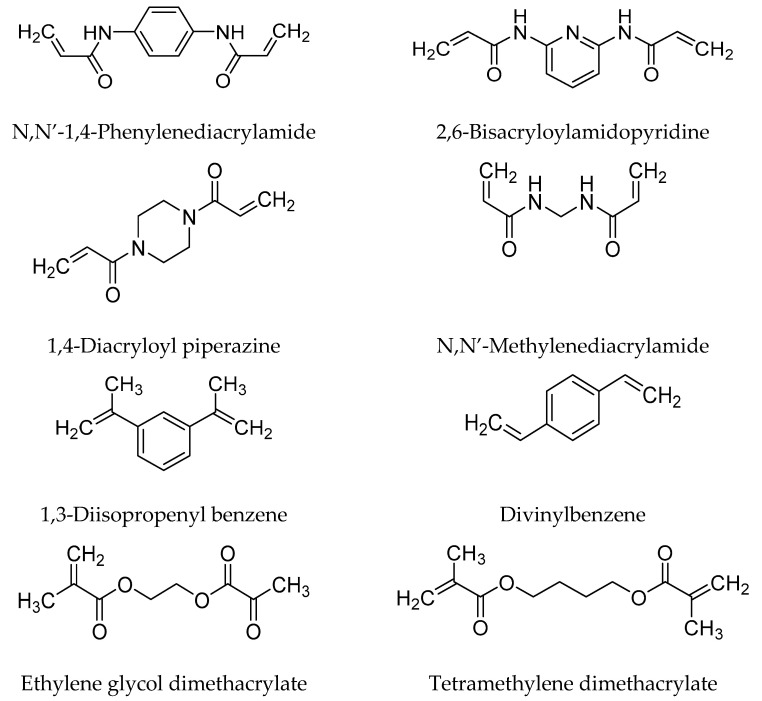
Chemical structures of commonly used cross-linking agents in molecular imprinting technique.

**Figure 6 polymers-14-00640-f006:**
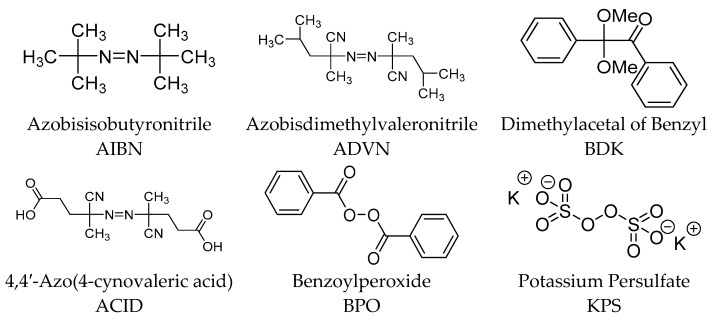
Structures of commonly used initiators in free radical polymerization.

**Figure 7 polymers-14-00640-f007:**
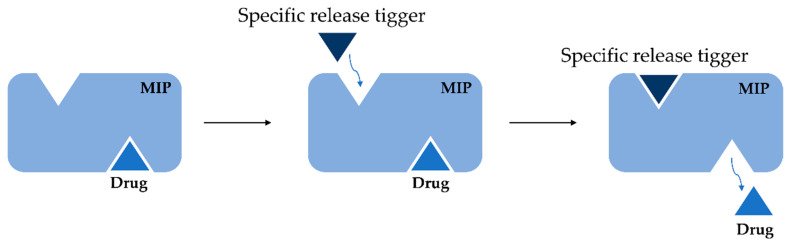
Scheme of intelligent drug release from MIP.

**Figure 8 polymers-14-00640-f008:**
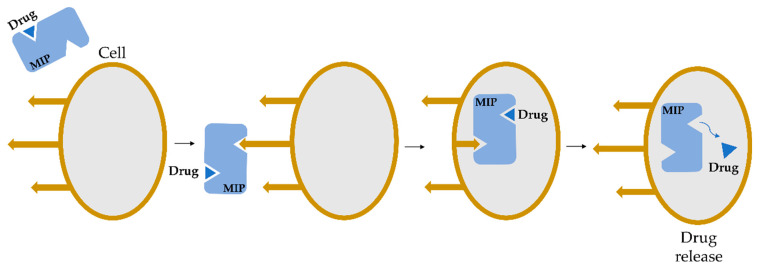
Scheme of targeted drug delivery into a cell with MIP.

**Figure 9 polymers-14-00640-f009:**
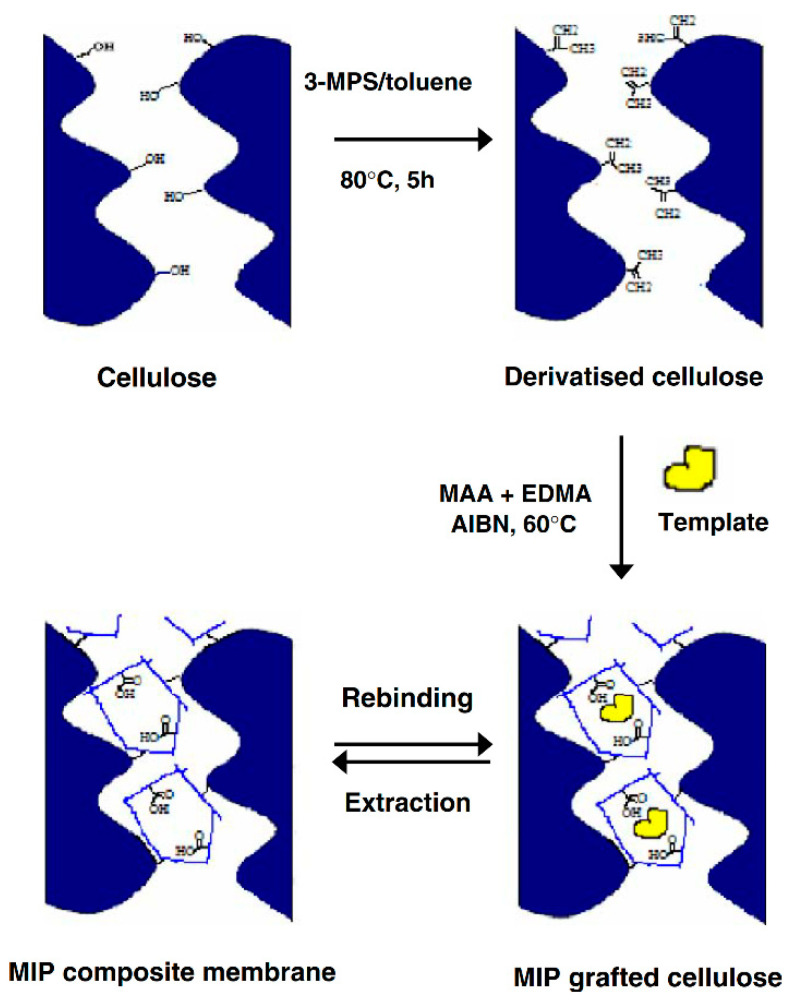
Scheme of cellulose membrane modified with a MIP. Reprinted from [37], Copyright (2022), with permission from Elsevier.

**Figure 10 polymers-14-00640-f010:**
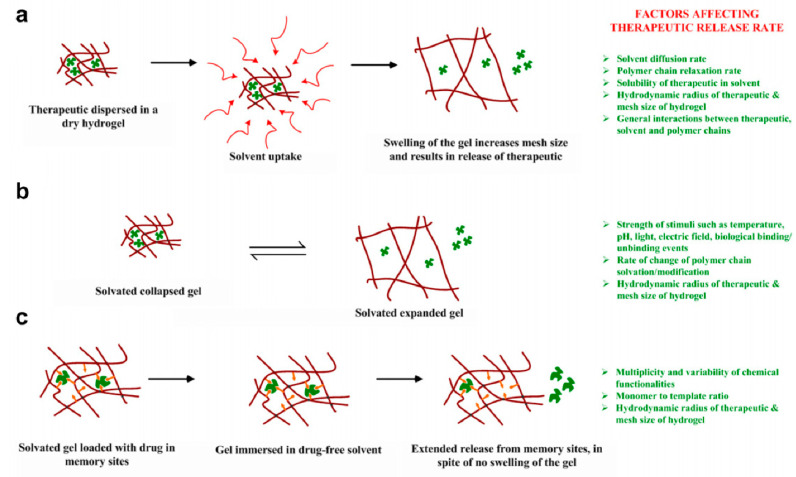
Scheme of (**a**) controlled drug release in hydrogels upon their swelling either by solvent uptake from a dry state or (**b**) thermodynamic compatibility with the solvent. (**c**) Macromolecular memory is obtained by imprinting a multifunctional pre-polymerization complex with the drug as an alternative strategy to release the drug in a controlled fashion when the gels are already solvated and fully swollen. Reprinted from [38], Copyright (2022), with permission from Elsevier.

**Figure 11 polymers-14-00640-f011:**
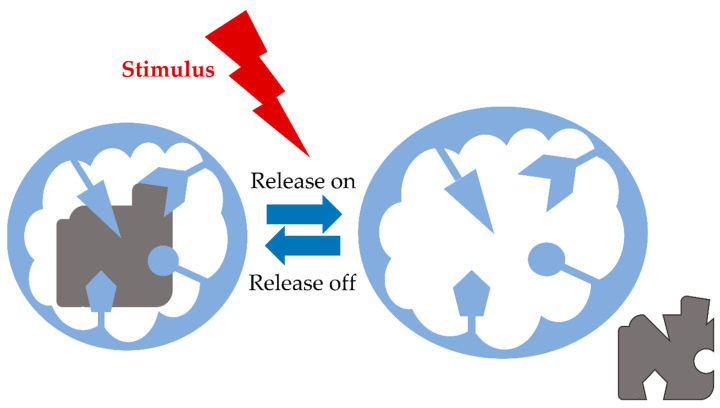
Schematic view of the effect of a stimulus on the conformation of the drug-imprinted cavities in a responsive hydrogel.

**Figure 12 polymers-14-00640-f012:**
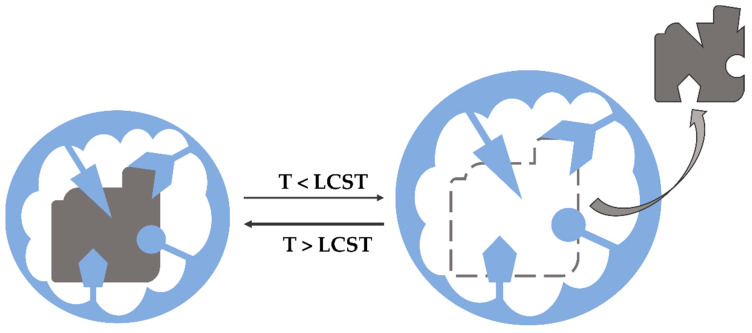
Scheme of the template’s removal mechanism from thermo-responsive MIPs.

**Figure 13 polymers-14-00640-f013:**
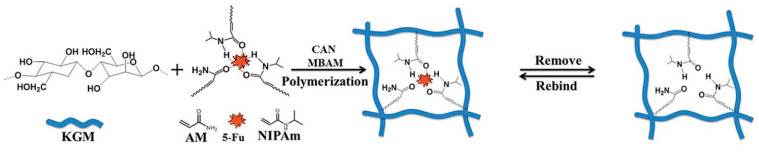
Schematic procedure of synthesis of 5-fluorouracil thermo- responsive MIP reported by Ann et al. Reprinted [69], Copyright (2022), with permission from Elsevier.

**Figure 14 polymers-14-00640-f014:**
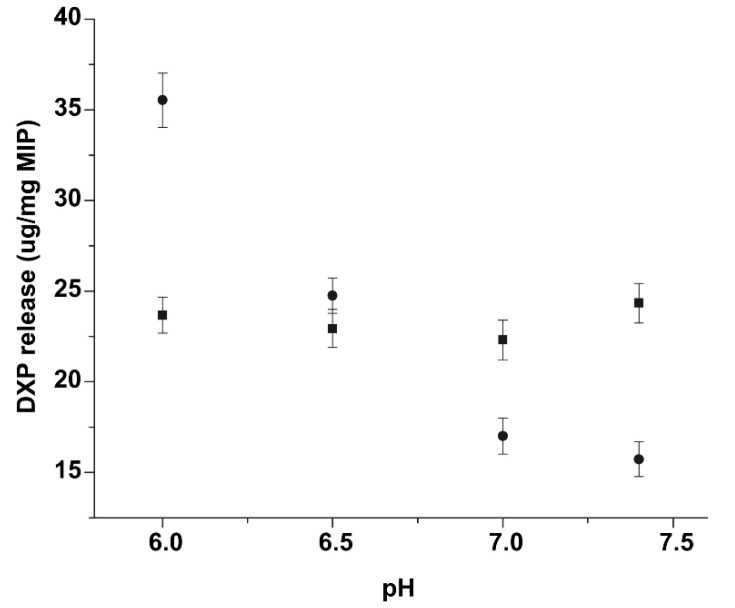
Scheme presenting DXP release profiles at various pH PBS solution at 37 °C [(▪) non-imprinted polymer; (●) imprinted polymer]. Reprinted from [72], Copyright (2022), with permission from Elsevier.

**Figure 15 polymers-14-00640-f015:**
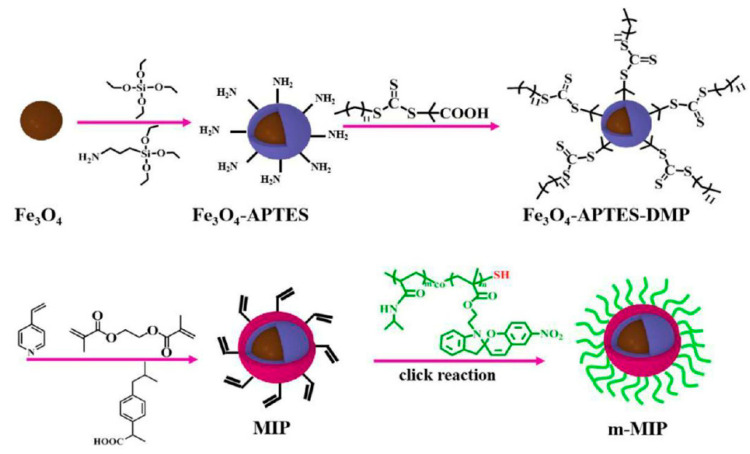
Schematic synthesis of core-shell MIP nanospheres with multiple responsive properties. Reprinted with permission from [81]. Copyright (2022) American Chemical Society.

**Table 1 polymers-14-00640-t001:** Examples of well-reported dual/multiple-responsive MIPs with their applications [16].

Type of Polymer	Template	Responsive Element	Application	Reference
Thermo/Magnetic	2,4,5-Trichlorophenol	NIPAAm, Fe_3_O_4_	Selective separation and enrichment fields	[74]
	Sulfamethazine	NIPAAm, *γ*-Fe_3_O_4_	Separation, drug release, protein recognition	[75]
	BSA	NIPAAm, Fe_3_O_4_	Chromatographic separation, solid-phase extraction, drug delivery. Medical diagnosis and biosensors	[76]
Photo/Magnetic	Caffeine	Fe_3_O_4_, MPABA	Trace caffeine analysis	[77]
pH/Thermo	Ovalbumin	NIPAAm, boronic acid	Chemical sensing and biosensing	[78]
Thermo/Photo	2,4-D	Azobenzene, NIPAAm	Separation, extraction, assays, drug delivery, and bioanalytical analysis	[79]
Thermo/Salt	BSA	NIPAAm, NaCl	Solid-phase extraction, sensors, and protein delivery agents	[80]
Thermo/Salt/Bio-molecule	Lysozyme or Cytochrome 4	NIPAAm, NaCl, Bio-molecule	Non-protein acetous receptor	[65]

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
