# Peer review of "Molecularly Imprinted Polymers as State-of-the-Art Drug Carriers in Hydrogel Transdermal Drug Delivery Applications"

_polymers, 2022, doi:10.3390/polym14030640_

Round 1
Reviewer 1 Report
The review manuscript entitled “Molecularly Imprinted Polymers as State-of-the-Art Drug Carriers in Hydrogel Transdermal Drug Delivery Applications” is quite interesting. Although many reviews targeting MIPs and their applications in several fields were published, the approached theme is original, as there are very few papers dealing with MIPs for transdermal drug delivery applications. Therefore, I recommend publication of this review in Polymers with some minor corrections, as follows:
- Since the target of this review is MIPs for transdermal drug delivery, I think the abstract should be better build in this direction. The introduction in MIP field is too wide to attract the attention towards the final aim.
- Pg 2-Line 54: “Following…” should be reformulated.
- Figures 1 and 2 should be merged and the difference of the three methods better highlighted in the scheme. The difference should be more obvious for the reader.
- Figure 10 is very blurry. You cannot distinguish the writing on the wright side of the figure. The resolution should be improved.
Author Response
Please see the attachemnt

Reviewer 2 Report
- What are the advantages and disadvantages of MIP compared with other polymers?
- Section 2.2. please mention all the routes that MIP has been applied for drug delivery in them and compare their release percentage in a table.
- Section 3 is just showing the challenges of transdermal delivery, how about the challenges of other routes?
- Figure 1 in covalent imprinting, one more step is required to show the addition of the crosslinker process.
- Why do the figure captions have references?
- Almost all the figures have very low quality and are not publishable. The texts in the figure are not readable and useless. Edit the figures, please.
- It is recommended to use the following reference in this manuscript:
Sabbagh, Farzaneh, and Beom Soo Kim. "Recent advances in polymeric transdermal drug delivery systems." Journal of Controlled Release 341 (2022): 132-146.
- The references older than 2010 are considered too old references. It is suggested to replace them with newly published papers.
Author Response
Please see the attachement
